# Aspartame and Phe-Containing Degradation Products in Soft Drinks across Europe

**DOI:** 10.3390/nu12061887

**Published:** 2020-06-24

**Authors:** Kimber van Vliet, Elise S. Melis, Pim de Blaauw, Esther van Dam, Ronald G. H. J. Maatman, David Abeln, Francjan J. van Spronsen, M. Rebecca Heiner-Fokkema

**Affiliations:** 1Division of Metabolic Diseases, University Medical Center Groningen, University of Groningen, P.O. Box 30.001, 9700 RB Groningen, The Netherlands; k.van.vliet@umcg.nl (K.v.V.); e.van.dam@umcg.nl (E.v.D.); f.j.van.spronsen@umcg.nl (F.J.v.S.); 2Laboratory of Metabolic Diseases, University Medical Center Groningen, University of Groningen, P.O. Box 30.001, 9700 RB Groningen, The Netherlands; esmelis26@hotmail.com (E.S.M.); p.de.blaauw@umcg.nl (P.d.B.); r.g.h.j.maatman@umcg.nl (R.G.H.J.M.); 3Dutch PKU Society, P.O. Box 91, 4000 AB Tiel, The Netherlands; david_abeln@yahoo.com

**Keywords:** aspartame, phenylalanine, aspartylphenylalanine, diketopiperazine, soft drinks, phenylketonuria, tyrosinemia type 1

## Abstract

Phenylketonuria and tyrosinemia type 1 are treated with dietary phenylalanine (Phe) restriction. Aspartame is a Phe-containing synthetic sweetener used in many products, including many ‘regular’ soft drinks. Its amount is (often) not declared; therefore, patients are advised not to consume aspartame-containing foods. This study aimed to determine the variation in aspartame concentrations and its Phe-containing degradation products in aspartame-containing soft drinks. For this, an LC–MS/MS method was developed for the analysis of aspartame, Phe, aspartylphenylalanine, and diketopiperazine in soft drinks. In total, 111 regularly used soft drinks from 10 European countries were analyzed. The method proved linear and had an inter-assay precision (CV%) below 5% for aspartame and higher CVs% of 4.4–49.6% for the degradation products, as many concentrations were at the limit of quantification. Aspartame and total Phe concentrations in the aspartame-containing soft drinks varied from 103 to 1790 µmol/L (30–527 mg/L) and from 119 to 2013 µmol/L (20–332 mg/L), respectively, and were highly variable among similar soft drinks bought in different countries. Since Phe concentrations between drinks and countries highly vary, we strongly advocate the declaration of the amount of aspartame on soft drink labels, as some drinks may be suitable for consumption by patients with Phe-restricted diets.

## 1. Introduction

Several inborn errors of metabolism require monitoring of daily dietary phenylalanine (Phe) intake. These include disorders causing hyperphenylalaninemia and disorders treated with 2-(2-nitro-4-trifluoromethylbenzoyl)-1,3-cyclohexanedione (NTBC). A well-known example of a disorder associated with hyperphenylalaninemia is phenylketonuria (PKU; McKusick 261600), caused by a deficiency of the enzyme phenylalanine hydroxylase [1,2]. This leads to a diminished or sometimes even blocked conversion of Phe into tyrosine (Tyr). The resulting high Phe concentrations in blood and brain are detrimental for neurological development [3]. Therefore, the treatment of PKU and some other hyperphenylalaninemic disorders largely consists in dietary Phe restriction. Disorders treated with NTBC, on the other hand, include tyrosinemia type 1 (TT1; McKusick 276700) and alkaptonuria (AKU; McKusick 203500) [4,5,6], which are both caused by a deficiency of an enzyme in the Tyr degradation pathway. NTBC blocks the activity of the upstream enzyme 4-hydroxyphenylpyruvate dioxygenase, thereby preventing the accumulation of toxic products in TT1 and AKU patients. This, however, leads to high Tyr concentrations, thereby necessitating a dietary restriction of Tyr and its precursor Phe. The dietary treatment of the mentioned diseases consists in a low dietary intake of natural proteins combined with an amino acid mixture devoid of Phe (and sometimes Tyr). Phe is present in variable amounts in different food products, with high Phe concentrations in meat, poultry, fish, eggs, soy products, dairy products, some vegetables, and, of interest, aspartame (APM).

APM is a synthetic sweetener approximately 100–200 times sweeter than sucrose [7]. Therefore, it is a low-caloric alternative for common sweeteners such as sucrose and is nowadays used in food products, particularly in soft drinks [8]. APM has been approved by the Food and Drug Administration (FDA) as a food additive (E-951), and the use of APM in products needs to be documented on the label. APM is a methylester of the dipeptide aspartate and Phe, and therefore a source of Phe. Patients with Phe-restricted diets are therefore recommended not to consume APM-containing soft drinks. Unfortunately, it is difficult to find specific information on the APM contents in soft drinks, as it is not mandatory for manufacturers to declare the amount of APM.

There are some publications on APM concentrations in a small number of soft drinks [9,10,11,12,13,14,15,16]. These studies analyzed APM mostly in addition to other sweeteners to investigate consumption or to study the instability of APM. APM is instable at high temperatures and at high and very low pH, which cause its degradation to other components, including 5-benzyl-3,6-dioxo-2-piperazine acetic acid [diketopiperazine (DKP)], aspartate, aspartylphenylalanine (Asp–Phe), and Phe [7,17]. Most of these degradation products are also sources of Phe. Patients with Phe-restricted diets would benefit from information on the amount of APM and other Phe-containing metabolites in soft drinks, as some soft drinks may be suitable as part of their diets.

This study aimed to investigate the concentrations of APM and most of its Phe-containing degradation products, including DKP, Asp–Phe, and Phe, in regularly used soft drinks from European countries. For this purpose, we developed a liquid chromatography–Tandem mass spectrometry (LC–MS/MS) method for the simultaneous quantitative analysis of APM and its metabolites.

## 2. Materials and Methods

### 2.1. Chemicals and Reagents

L-APM was obtained from the United States Pharmacopeia (Basel, Switzerland), L-APM-d5 from Toronto Research Chemicals (North York, ON, Canada), aspartic acid, aspartylphenylalanine, Phe, and diketopiperazine were obtained from Sigma-Aldrich Corporation (Saint Louis, MO, USA). Methanol ≥99.98% and formic acid 99% were obtained from Biosolve (Valkenswaard, The Netherlands), and HCl 37% from Merck Darmstadt.

### 2.2. Samples

Contact persons in 10 European countries were approached to send us bottles or cans of soft drinks, obtained from local supermarkets. To study within- and between-batch variation, three samples from three batches of Coca Cola zero and Fanta Orange zero (i.e., 9 samples, 3 batches per soft drink) were bought in The Netherlands and were analyzed in duplicate.

### 2.3. Analyses

Calibration solutions, quality control samples (4 selected soft drinks), and soft drinks (2 mL) were de-gassed for 20 min in an ultrasonic bath at 10 °C. Then, 10 µL of either sample or calibration solution was subsequently added to 1490 µL of internal standard solution (6.7 µM APM-D5 in 0.1 M HCl) and vortexed for 10 s. For the analysis, 5 µL of this sample was injected onto a high-performance liquid chromatography (HPLC) column. All samples were analyzed in duplicate using a HPLC (LC20; Shimadzu, Kyoto, Japan) coupled to a triple quadrupole mass spectrometer with an electrospray ionization source (API-3200, SCIEX, Framingham, MA, USA). LC–MS/MS analysis was carried out using a Kinetex Biphenyl (2.6 µm pore size, 150 × 4.6 mm) analytical column coupled to a Kinetex Biphenyl (2.6 µm pore size, 2 × 4.6 mm) guard column (Phenomenex, Torrance, CA, USA). Separation was achieved in 5 min by applying isocratic elution (40% mobile phase A: 0.1% formic acid in MilliQ water and 60% mobile phase B: 100% methanol), at a flow rate of 0.6 mL/min and a column temperature of 40 °C. Detection was achieved using positive-ion electrospray ionization in multiple reaction monitoring mode, using the following transitions: m/z 295.2 > 235.3 for APM, m/z 300.2 > 125.2 for D5-APM, m/z 263.1 > 91.2 for DKP, m/z 281.2 > 166.3 for Asp–Phe, and m/z 166.1 > 120.2 for Phe. The electrospray ionization source temperature was kept at 450 °C, the ion spray voltage at 4000 V, and nitrogen was used as the nebulizing gas. Data were analyzed using Analyst 1.6.2 (Sciex, Framingham, MA, USA).

### 2.4. Validation

Four soft drinks with different amounts of the various metabolites were selected for precision experiments. Intra-day precision was calculated from 10 replicates analyzed in a single analytical run. Inter-assay variation was calculated from the samples analyzed in duplicate on 7 different days. Precision was calculated as %CV = standard deviation/mean of replicates.

For linearity, an eight-point calibration curve in the concentration range of 0.00–3.45 µmol/L for APM, 0.00–0.903 µmol/L for Phe, 0.00–0.576 µmol/L for DKP, and 0.00–0.746 µmol/L for Asp–Phe dissolved in 0.1 M HCl was analyzed 7 times, including 1 time in random order and 1 time in high–low order. Slope (K1), correlation coefficient (R), and CV were calculated to determine the linearity. In addition, the calibration samples were diluted in two soft drinks to determine the effect of the soft drink matrix on the analysis. The matrix effect and mean recovery were investigated by comparing the slopes of the calibration curves, and a difference <15% was considered acceptable. The difference is also the representative of the mean recovery of metabolites from the matrix.

The limit of detection (LOD) and the limit of quantification (LOQ) were calculated from the 7 calibration curves. The standard deviation of the y-intercepts and the mean slope of the regression models were calculated. The LOD was defined as (3.3 × sd)/slope, whereas the LOQ was defined as (10 × sd)/slope [18]. Carry over was determined by analyzing QC high (H) and QC low (L) 10 times in a specific order; L1 L2 H1 H2 L3 H3 H4 L4 L5 L6 L7 H5 H6 L8 H7 H8 L9 H9 H10 L10 (EP Evaluator 12.0, Data Innovations LLC, South Burlington, Vermont, USA). The mean and standard deviation of low-after-low (LL: L2, L5, L6 and L7) and low-after-high (LH: L3, L4, L8, L9 and L10) were calculated. Carry over was considered present when mean LH − mean LL <3 × SD of LL.

The stability of APM and its metabolites was determined in duplicate in three soft drinks after 0, 1, 2, 3, 7, and 14 days of storage at ±37 °C, ±22 °C, ±2 °C, and ±−20 °C. Moreover, the stability of the processed samples was assessed after 0, 2, 4, 8, 16, and 24 h of storage in the autosampler at 15 °C. A deviation >15% from t = 0 was considered significant.

### 2.5. Statistics

The contents of APM and its metabolites were expressed in µmol/L. The total Phe amount in soft drinks was calculated from the concentrations of APM and Phe-containing metabolites, as 1 mol of APM or Phe-metabolite is equal to 1 mol of Phe. Within- and between-batch variation was calculated as coefficients of variation (CV; standard deviation/mean) in percentage. In addition to this, differences between batches were tested using Kruskal–Wallis tests. Statistical analyses were performed using IBM SPSS Statistics 23rd version, and *p*-values < 0.05 were considered statistically significant.

## 3. Results

### 3.1. Method Validation

The inter- and intra-assay precision of the newly developed method for the analysis of APM and its degradation products were determined by analyzing four soft drinks, see Table 1. The method proved linear in the concentration range of 0–3450 µmol/L for APM (R = 0.9995, CV = 0.03%), 0–900 µmol/L for Phe (R = 0.9998, CV = 0.01%), 0–576 µmol/L for DKP (R = 0.9994, CV = 0.02%), and 0–746 µmol/L for Asp–Phe (R = 0.9996, CV = 0.01%). The LOD/LOQ were 0.002/0.007 µmol/L for APM, 10/30 µmol/L for Phe, 7/22 µmol/L for DKP, and 17/50 µmol/L for Asp–Phe. There was no carry-over (see Appendix A). There was no matrix effect for the four components, the slopes of the calibration standards diluted in soft drinks A and B were comparable (i.e., less than 15% different) to those of the standards diluted in 0.1 M HCl, i.e., 94% (APM), 100% (Phe), 91% (DKP), and 110% (Asp/Phe) for soft drink A and 96% (APM), 96% (Phe), 91% (DKP), and 101% (Asp/Phe) for soft drink B. These results also represent the mean recoveries of the components. Results of the stability experiment are shown in Table 2. Aspartame concentration considerably decreased when heated up to ±37 °C, whereas the concentrations of the degradation products DKP and Asp–Phe increased at this temperature. The total Phe concentration also decreased, but to a lower extent. The concentrations of aspartame and total Phe were relatively stable at all other temperatures. DKP and to a lesser extent Asp–Phe concentrations increased when the samples were kept heated up to ±37 °C or at room temperature. DKP concentration increased in S-2 also when stored cooled or frozen until day 6, whereas no changes occurred in the more concentrated sample (S-3) at these lower temperatures.

### 3.2. Aspartame and its Metabolites in Soft Drinks across Europe

A total of 111 soft drinks in original cans or bottles were obtained from Belgium (*n* = 15), Denmark (*n* = 2), Finland (*n* = 21), France (*n* = 4), Germany (*n* = 7), Spain (*n* = 7), Sweden (*n* = 4), The Netherlands (*n* = 38), Turkey (*n* = 4), and the United Kingdom (*n* = 9). Sixteen soft drinks did not contain APM according to their labels. APM and APM degradation products were measured using our validated method. The results of APM and total Phe are shown in Figure 1. APM and total Phe were not detectable in the 16 non-APM soft drinks.

The 111 soft drinks were divided into four subgroups; group A (orange drinks), group B (lemon drinks), group C (cola drinks), and group D (other drinks). Similar drinks, such as diet, sugar-free, or flavored versions, were clustered. APM, Phe, DKP, Asp–Phe, and total Phe concentrations for the different groups are shown in Appendix A. The percentage of Phe that derived from non-APM products amounted to a mean of 14% of total Phe, ranging from 4 to 51%.

### 3.3. Within- and Between-Batch Variation (Same Country)

Within-batch variations were 2.7, 2.6, 3.3% (APM) and 2.6, 2.7, 3.0% (total Phe) for the three batches of Fanta Orange zero and 3.6, 1.7, 4.8% (APM) and 3.7, 1.6, 4.6% (total Phe) for the three batches of Coca Cola zero. Figure 2 shows the variation in APM between the batches. Between-batch CV were 3.8% (APM) and 4.2% (total Phe) for Fanta Orange zero and 3.6% (APM) and 3.4% (total Phe) for Coca Cola zero. Kruskal–Wallis tests showed small, but significant differences in APM and total Phe concentrations between the three batches of Fanta orange zero (*p* = 0.010 and *p* = 0.013). No significant differences were observed between the three Coca Cola zero batches. Finally, there were no significant differences between samples from the same batch.

### 3.4. Between-Batch Variation between Countries

Results for APM, degradation products, and total Phe of all soft drinks are shown in Appendix A. Large between-country variations were observed.

## 4. Discussion

This study developed and validated a method for measuring APM and its degradation products, including Phe, in soft drinks using LC–MS/MS. This method was used to measure APM and its degradation products in regularly used soft drinks across Europe. We observed a large variation in the amount of APM and its degradation products in the measured soft drinks. Also, substantial differences were observed in the concentrations measured in similar soft drinks bought in different countries.

The mainstay of Phe-restricted diets is limiting the daily natural protein intake. One gram of natural protein is considered to equal approximately 50 mg (or 303 µmol) of Phe [19]. Protein tolerance, however, can differ significantly among patients, depending on, e.g., residual enzyme activity, pregnancy, age, weight, and for hyperphenylalaninemias, also tetrahydrobiopterin (BH4) responsiveness [1]. Healthy adults are used to consume approximately 70–120 g of natural protein a day, whereas patients with a Phe-restricted diet are recommended to only consume 5–20 g of natural protein, which is approximately equal to 250–1000 mg (or 1513–6054 µmol) Phe a day. Because of their dietary Phe restriction, patients are told not to consume products containing APM. The results of this study showed, however, a wide variety of APM and total Phe concentrations in soft drinks. The APM and total Phe concentrations in our 95 APM-containing drinks ranged from 103 to 1790 µmol/L (30–527 mg/L) and from 119 to 2013 µmol/L (20–332 mg/L), respectively. APM concentrations were to some extent comparable to concentrations reported earlier [9,10,11,12,13,14,15,16], except those reported by Zhu et al., who found significantly higher APM concentrations of 2826 and 7235 mg/L in two soft drinks [10]. However, none of the earlier studies measured as many soft drinks as our study did, nor similar soft drinks of the same company from batches obtained from different countries. Sakai et al. also measured DKP in three soft drinks, ranging from 1.6 to 2.9 mg/L, which is comparable to concentrations found in our study [9]. The comparability of our results with those of other studies confirms the validity of our data.

Including APM breakdown products is in our opinion essential to estimate the total original APM amounts, as up to 51% of total Phe derived from non-APM Phe in the investigated soft drinks. This is supported by the results of Barrado et al., who showed a very high Phe/APM ratio in two soft drinks [13]. Analysis of only APM may therefore (highly) underestimate the total Phe content. Also of interest is that APM concentrations were not detected in 16 non-APM-containing soft drinks, although some did contain very small amounts of total Phe ranging from 0 to 12.5 µmol/L. In patients with the lowest protein tolerance (5 g natural protein per day), consumption of one glass (200 mL) of an APM-containing soft drink would correspond approximately to 0.4–26.4% of the daily allowable Phe intake. Some APM-containing soft drinks should therefore indeed be seen as undesirable, but others can be considered to have safe Phe concentrations for consumption by patients with milder deficiencies. Please note that the amount of APM in the soft drinks presented in Appendix A should be taken with caution, as it is not clear whether and when manufacturers change their recipes. Furthermore, it is important to note here that, mainly because of the high carbohydrate content, the consumption of soft drinks in general is not considered to be part of a healthy diet.

Since the FDA approval for APM many years ago, the safety of APM has been studied extensively. Numerous articles have been written regarding possible toxic, mutagenic, and carcinogenic effects [20]. Although the widespread debate on APM safety is ongoing, studies have been extensively reviewed by the European Food Safety Authority (EFSA), who found no significant toxic effects and who recommend dosages <40 mg/kg/day for non-PKU individuals [8]. Not only soft drinks, but also other food products and several medications contain APM [8]. We did not investigate APM in other food products or medications. However, as shown in this study, patients using Phe-restricted diets may be able to consume some of the APM-containing soft drinks. In addition, it is possible that they will also be able to safely consume some of the APM-containing foods and medicines if the amount of APM is reported on the label. Of note, physicians should not hesitate to treat patients with APM-containing medications, when this is considered necessary.

A limitation of this study is that we did not ask our representatives from different countries for specific soft drinks. This would have improved the comparison of the APM amounts in soft drinks between countries. Instead, we asked to send us the most regularly consumed soft drinks, as applicability for patients was the main goal of our study. Furthermore, APM is unstable at high temperatures and at low and high pH. The effect of short-term storage at 37 °C was also apparent from the results of this study. The exact storage and transportation conditions of the samples before they arrived at our laboratory are unknown and may have affected the APM concentrations, especially if the samples were sent from warmer South European countries. However, by also measuring the degradation products, the total Phe content in the soft drinks is expected to better reflect the added APM amount. However, the total Phe concentration also decreased at 37 °C, although to a lesser extent than that of APM. Our study did not include two other breakdown products of APM, i.e., Phe-methylester and phenylalanylaspartate, which may also be sources of Phe in soft drinks. Their contribution to total Phe is yet unknown but not likely to contribute to a higher extent compared to the metabolites tested. The amount contributed by Phe-methylester seems to be very limited [7].

Another limitation of our study is that our method was designed primarily to analyze APM. The measurement of the degradation products was a second aim. To analyze all metabolites in a single run, higher limits of quantification for the degradation products were tolerated. In many samples, these metabolites had concentrations below the LOQ (Phe 100%, DKP 40%, Asp–Phe 51%).

## 5. Conclusions

Our laboratory developed and validated a new method for measuring APM and its degradation products in soft drinks. Regularly used soft drinks across Europe were shown to have varying amounts of APM and Phe, even in comparable (clustered) soft drinks. Patients with hyperphenylalaninemias and patients treated with NTBC can only consume limited amounts of Phe during the day. Since some soft drinks had only low amounts of APM and Phe, patients should be able to consume these drinks. Therefore, we strongly recommend national and international patient societies and advocacy groups to call for the responsibility of companies to declare the amount of APM in their soft drinks.

## Figures and Tables

**Figure 1 nutrients-12-01887-f001:**
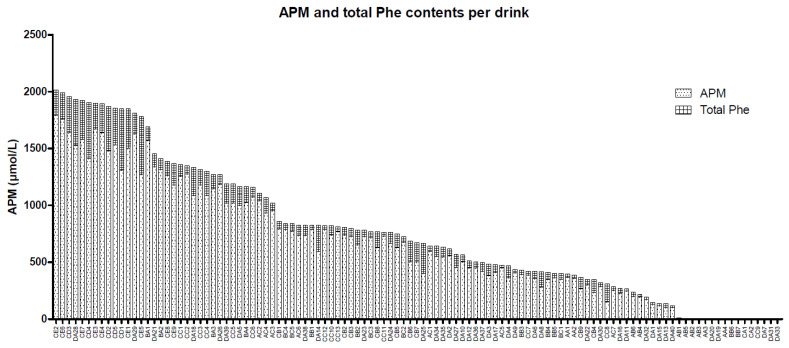
Aspartame and total phenylalanine contents of all 111 soft drinks. Specifics regarding the soft drinks and their countries of origin can be found in Appendix A, where the first letter indicates the group, the second letter indicates the cluster, and the number indicates the number of the soft drink. APM, aspartame, Phe, phenylalanine.

**Figure 2 nutrients-12-01887-f002:**
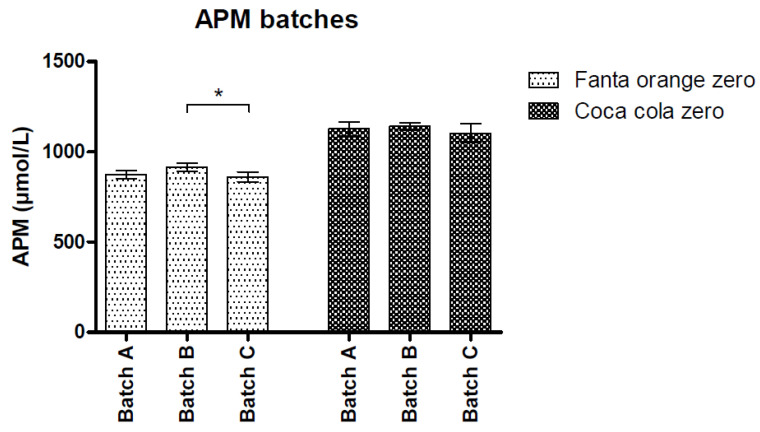
Between-batch variation of aspartame and total phenylalanine concentrations for Fanta orange zero and Coca cola zero sugar; * *p*-value < 0.05. Data represent mean ± SD.

**Table 1 nutrients-12-01887-t001:** Intra- and inter-assay precision of the method for the analysis of aspartame and its degradation products.

Sample		APM	Phe	DKP	Asp–Phe	Total Phe
		CV_a_ intra-assay				
Soft drink A *N* = *10*	Mean (µmol/L)	349.5	5.5	22.4	23.1	378.6
	CV (%)	3.3%	8.1%	6.8%	1.8%	3.1%
Soft drink B *N* = *10*	Mean (µmol/L)	1559.0	12.3	30.4	25.1	1601.7
	CV (%)	2.5%	3.7	7.6%	1.4%	2.4%
Soft drink C *N* = *10*	Mean (µmol/L)	2747.0	16.1	141.2	72.9	2878.5
	CV (%)	3.8%	7.3%	3.3%	3.1%	3.8%
Soft drink D *N* = *10*	Mean (µmol/L)	634.2	9.7	86.4	86.2	816.5
	CV (%)	2.7%	4.1%	2.9%	3.8%	2.9%
		CV_a_ inter-assay				
Soft drink A *N* = *14*	Mean (µmol/L)	203.4	3.9	10.3	21.1	238.1
	CV (%)	4.9%	38.6%	33.6%	14.4%	5.4%
Soft drink B *N* = *14*	Mean (µmol/L)	920.9	11.1	26.7	36.8	995.5
	CV (%)	4.7%	23.8%	17.4%	12.0%	4.7%
Soft drink C *N* = *14*	Mean (µmol/L)	1664.3	4.1	77.0	139.2	1884.6
	CV (%)	3.9%	49.6%	22.9%	11.9%	4.4%
Soft drink D *N* = *14*	Mean (µmol/L)	622.6	13.3	100.7	104.8	841.3
	CV (%)	3.8%	22.3%	16.9%	11.4%	4.7%

APM, aspartame, Phe, phenylalanine, DKP, diketopiperazine, Asp–Phe, aspartylphenylalanine, CV, coefficient of variation.

**Table 2 nutrients-12-01887-t002:** Stability experiment.

		APM			Phe			DKP			Asp–Phe			Total Phe		
		S-1	S-2	S-3	S-1	S-2	S-3	S-1	S-2	S-3	S-1	S-2	S-3	S-1	S-2	S-3
Reference (µmol/L)	Day 0	227.0	1008.0	1880.0	1.30 (<LOD)	7.30 (<LOD)	0.99 (<LOD)	0.00 (<LOD)	3.18 (<LOD)	54.1	22.9	24.3	121.0	249.9	1035.5	2055.1
Heated ±37 °C	Day 1	100%	100%	98%	<LOD	<LOD	<LOD	<LOD	<LOD	**129%**	108%	112%	109%	101%	100%	100%
	Day 6	**79%**	**77%**	**78%**	<LOD	<LOD	<LOD	<LOD	<LOD	**214%**	96%	**149%**	**131%**	87%	**83%**	**85%**
	Day 15	**64%**	**64%**	**69%**	<LOD	<LOD	<LOD	<LOD	<LOD	**375%**	**120%**	**217%**	**181%**	**79%**	**74%**	**84%**
Room temp ±22 °C	Day 1	95%	99%	93%	<LOD	<LOD	<LOD	<LOD	<LOD	101%	101%	114%	102%	96%	100%	94%
	Day 6	104%	93%	90%	<LOD	<LOD	<LOD	<LOD	<LOD	**124%**	**124%**	105%	103%	112%	95%	92%
	Day 15	86%	93%	90%	<LOD	<LOD	<LOD	<LOD	<LOD	**153%**	98%	**146%**	**128%**	90%	96%	94%
Cooled ±2 °C	Day 1	100%	99%	95%	<LOD	<LOD	<LOD	<LOD	<LOD	101%	105%	110%	98%	101%	99%	95%
	Day 6	97%	97%	93%	<LOD	<LOD	<LOD	<LOD	<LOD	106%	**79%**	91%	95%	98%	98%	93%
	Day 15	94%	90%	95%	<LOD	<LOD	<LOD	<LOD	<LOD	108%	86%	103%	102%	95%	91%	96%
Frozen ±−20 °C	Day 1	93%	98%	98%	<LOD	<LOD	<LOD	<LOD	<LOD	102%	95%	106%	102%	93%	98%	98%
	Day 6	100%	94%	96%	<LOD	<LOD	<LOD	<LOD	<LOD	105%	**80%**	87%	93%	101%	94%	96%
	Day 15	95%	93%	96%	<LOD	<LOD	<LOD	<LOD	<LOD	109%	86%	96%	96%	96%	94%	97%

S-1, S-2, and S-3 are the different soft drinks used in the stability experiment. Concentrations are expressed as a percentage relative to the concentration on day 0. Deviations of more than 15%, shown underlined and in bold, were considered significant. Concentrations were below the limit of detection (LOD) in all samples for Phe (<10 µmol/L) and in samples 1 and 2 for DKP (<7 µmol/L) and therefore not suitable for investigation of (in)stability.

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
