# Peer review of "Aspartame and Phe-Containing Degradation Products in Soft Drinks across Europe"

_nutrients, 2020, doi:10.3390/nu12061887_

Round 1
Reviewer 1 Report
This is a very novel and interesting paper highlighting an important aspect of dietary management of patients with inborn errors of phenylalanine and tyrosine.
The paper is well written paper, reading well, and concluding that labeling of these drinks would be important in order to allow its consumption by the patients with these rare diseases. I completely agree with this statement.
Despite this, on page 9, line 222, the authors state that these products may be considered safe. Although it is understandable that they were referring to the total phenylalanine content, the authors should underline that, independently of the content in aspartame and its metabolites, these drinks should not be ingested without criteria. It would be important to underline that moderate consumption is recommended considering the potential negative impact on gut microbiota, increasing the risk for some of the features of the metabolic syndrome. It is not the case here, but in all processed foods, the use of additives is not of benefit. For example, the impact of emulsifiers is linked with TMA production by the gut microbiota, increasing TMAO levels and the cardiovascular risk. This topic can be reviewed here: Zmora N, Suez J, Elinav E. You are what you eat: diet, health and the gut microbiota. Nat Rev Gastroenterol Hepatol. 2019;16(1):35‐56. doi:10.1038/s41575-018-0061-2.
For the sweeteners I would recommend including the following publication which highlight the link between non-nutritive sweeteners and metabolic syndrome: Liauchonak I, Qorri B, Dawoud F, Riat Y, Szewczuk MR. Non-Nutritive Sweeteners and Their Implications on the Development of Metabolic Syndrome. Nutrients. 2019;11(3):644. Published 2019 Mar 16. doi:10.3390/nu11030644.
The authors should also make a clinical link of their results, focusing a preventive approach. There are recent papers underlining the risk of insulin resistance in patients suffering from phenylketonuria and we should not rule out the possibility of a gut microbiota induced effect. In fact, recent evidence demonstrates the possibility of dysbiosis in phenylketonuria (Bassanini G, Ceccarani C, Borgo F, et al. Phenylketonuria Diet Promotes Shifts in Firmicutes Populations. Front Cell Infect Microbiol. 2019;9:101. Published 2019 Apr 16. doi:10.3389/fcimb.2019.00101). In this case, it should be noted that referring only “safe ingestion” of these drinks may not be completely prudent since it may increase its consumption leading to a maintained dysbiosis with potential negative long-term outcomes.
If these target-patients already have a carbohydrate rich diet it would be important to underline that sustained ingestion of these soft drinks (even with limited phenylalanine content) may be contributing to the addition effect to sweet foods ingestion increasing the risk for comorbidities.
Author Response
Reviewer 1
This is a very novel and interesting paper highlighting an important aspect of dietary management of patients with inborn errors of phenylalanine and tyrosine.
The paper is well written paper, reading well, and concluding that labeling of these drinks would be important in order to allow its consumption by the patients with these rare diseases. I completely agree with this statement.
Despite this, on page 9, line 222, the authors state that these products may be considered safe. Although it is understandable that they were referring to the total phenylalanine content, the authors should underline that, independently of the content in aspartame and its metabolites, these drinks should not be ingested without criteria. It would be important to underline that moderate consumption is recommended considering the potential negative impact on gut microbiota, increasing the risk for some of the features of the metabolic syndrome. It is not the case here, but in all processed foods, the use of additives is not of benefit. For example, the impact of emulsifiers is linked with TMA production by the gut microbiota, increasing TMAO levels and the cardiovascular risk. This topic can be reviewed here: Zmora N, Suez J, Elinav E. You are what you eat: diet, health and the gut microbiota. Nat Rev Gastroenterol Hepatol. 2019;16(1):35‐56. doi:10.1038/s41575-018-0061-2.
We would like to thank the reviewer for this comment. It is correct that when we stated that the products may be safe, we were referring to the phenylalanine contents of the drinks. On page 9, line 225 we have clarified this by stating that drinks can be considered to have safe Phe concentrations for consumption by patients with milder deficiencies.
We also agree with the reviewer that moderate consumption should still be recommended because of its potential negative impacts. Therefore, we included the following sentence on line 228-229: Furthermore, it is important to note here that, mainly because of its carbohydrate contents, consumption of soft drinks in general are not considered to be part of a healthy diet.
For the sweeteners I would recommend including the following publication which highlight the link between non-nutritive sweeteners and metabolic syndrome: Liauchonak I, Qorri B, Dawoud F, Riat Y, Szewczuk MR. Non-Nutritive Sweeteners and Their Implications on the Development of Metabolic Syndrome. Nutrients. 2019;11(3):644. Published 2019 Mar 16. doi:10.3390/nu11030644.
Thank you for this comment and the interesting reference. We have included this reference along with the following text on line 234-237: In this respect it is interesting to note that soft drinks with non-nutritive sweeteners, such as APM, may in large amounts increase the risk for developing the metabolic syndrome by changing the gut microbiota.
The authors should also make a clinical link of their results, focusing a preventive approach. There are recent papers underlining the risk of insulin resistance in patients suffering from phenylketonuria and we should not rule out the possibility of a gut microbiota induced effect. In fact, recent evidence demonstrates the possibility of dysbiosis in phenylketonuria (Bassanini G, Ceccarani C, Borgo F, et al. Phenylketonuria Diet Promotes Shifts in Firmicutes Populations. Front Cell Infect Microbiol. 2019;9:101. Published 2019 Apr 16. doi:10.3389/fcimb.2019.00101). In this case, it should be noted that referring only “safe ingestion” of these drinks may not be completely prudent since it may increase its consumption leading to a maintained dysbiosis with potential negative long-term outcomes.
We thank the reviewer for sharing his thoughts on PKU and the microbiome, in particular the potential dysbiosis, its possible negative long term outcomes and the negative impact of the Phe diet. However, as our manuscript has a primary focus on aspartame in soft drinks, we decided not to put too much focus on other aspects of PKU or its diet, although we acknowledge that this is a very interesting subject. We hope that the reviewer agrees with our approach.
If these target-patients already have a carbohydrate rich diet it would be important to underline that sustained ingestion of these soft drinks (even with limited phenylalanine content) may be contributing to the addition effect to sweet foods ingestion increasing the risk for comorbidities.
We thank the reviewer for this suggestion. We have added the following sentence (line 228-229) on the fact that soft drink intake is not considered to be part of a healthy diet in general: Furthermore, it is important to note here that, mainly because of its carbohydrate contents, consumption of soft drinks in general are not considered to be part of a healthy diet.
Reviewer 2 Report
This manuscript addresses an important issue in the context of PKU since aspartame is a source of Phe, which is highly avoided in this disease. The authors emphasize the lack of labelling regarding the amount of aspartame and advocate the declaration of this amount on labels, helping the professionals treating disorders like PKU and Tyrosinemia type 1 as well as patients.
I only have minor revisions with few notes that can improve the design of the paper.
Minor revisions
- Abstract
I suggest using Phe after the first-time authors introduce phenylalanine
- Introduction
Lines 37 and 38 - Replace “The resulting high phenylalanine concentrations…” by “The resulting high Phe concentrations…” because authors already introduced the abbreviation
Line 62 - Replace “(diketopiperazine (DKP))…” by “[diketopiperazine (DKP)]…” - square and round brackets
Line 68 - I suggest putting the meaning of LC-MS/MS because it is the first-time authors introduce the abbreviation (and in lines 89 and 90 just put the abbreviation)
- Material and methods
Lines 84, 94 and 97 - Replace “oC” by “ºC”
Lines 86 and 87 - I suggest putting the meaning of HPLC in line 86 because it is the first-time authors introduce the abbreviation and in line 87 just put the abbreviation
Line 130 - Please replace “23nd” by “23rd or version 23”
- Results
Table 1 - CVa intra-assay and CVa inter-assay - the letters are at different sizes
Figure 1 - I suggest putting Phe instead of phe in the graph
- References
References numbers are in duplicate:
- [1]
Author Response
This manuscript addresses an important issue in the context of PKU since aspartame is a source of Phe, which is highly avoided in this disease. The authors emphasize the lack of labelling regarding the amount of aspartame and advocate the declaration of this amount on labels, helping the professionals treating disorders like PKU and Tyrosinemia type 1 as well as patients.
I only have minor revisions with few notes that can improve the design of the paper.
Minor revisions
Abstract
- I suggest using Phe after the first-time authors introduce phenylalanine
Thank you for this suggestion. We have changed ‘phenylalanine’ to ‘Phe’ and included this abbreviation after the first mention of phenylalanine.
Introduction
- Lines 37 and 38 - Replace “The resulting high phenylalanine concentrations…” by “The resulting high Phe concentrations…” because authors already introduced the abbreviation
- Line 62 - Replace “(diketopiperazine (DKP))…” by “[diketopiperazine (DKP)]…” - square and round brackets
- Line 68 - I suggest putting the meaning of LC-MS/MS because it is the first-time authors introduce the abbreviation (and in lines 89 and 90 just put the abbreviation)
We thank the reviewer for these suggestions. We have changed phenylalanine into the abbreviation and have altered the brackets. In line 68 we have included the meaning Liquid Chromatography Tandem Mass Spectrometry.
Material and methods
- Lines 84, 94 and 97 - Replace “oC” by “ºC”
- Lines 86 and 87 - I suggest putting the meaning of HPLC in line 86 because it is the first-time authors introduce the abbreviation and in line 87 just put the abbreviation
- Line 130 - Please replace “23nd” by “23rd or version 23”
Thank you for your comments. We have replaced oC by °C. We have included the meaning of HPLC and the abbreviation at the first mention in the text and only used the abbreviation later on in the text. In line 130 we replaced 23nd with 23rd.
Results
- Table 1 - CVa intra-assay and CVa inter-assay - the letters are at different sizes
- Figure 1 - I suggest putting Phe instead of phe in the graph
Thank you for these comments. We have altered this to ensure the letters are the same sizes and we have changed phe in the Figure to Phe.
References
- References numbers are in duplicate:
1.[1]
Thank you for this comment. We have deleted the second reference numbers.
Reviewer 3 Report
Good paper - solid science - well done.
It would be nice to follow up with concentrations in blood of Phe for heavy users of aspartame sweetened soft drinks.
Author Response
Good paper - solid science - well done.
It would be nice to follow up with concentrations in blood of Phe for heavy users of aspartame sweetened soft drinks.
We thank the reviewer for the comment. We agree with the reviewer that a study on the blood Phe levels of heavy aspartame users would be a very interesting study. However, since our research focuses on the treatment of Phenylketonuria and Tyrosinemia type 1 patients who are currently strongly discouraged from using aspartame containing drinks, this study would be difficult for us to perform in our patient groups.